# Objective Olfactory Findings in Hospitalized Severe COVID-19 Patients

**DOI:** 10.3390/pathogens9080627

**Published:** 2020-07-31

**Authors:** Jerome R. Lechien, Morgane Ducarme, Sammy Place, Carlos M. Chiesa-Estomba, Mohamad Khalife, Giacomo De Riu, Luigi Angelo Vaira, Christophe de Terwangne, Shahram Machayekhi, Arnaud Marchant, Fabrice Journe, Sven Saussez

**Affiliations:** 1COVID-19 Task Force of the Young-Otolaryngologists of the International Federations of Oto-Rhino-Laryngological Societies (YO-IFOS), F92150 Paris, France; jerome.lechien@umons.ac.be (J.R.L.); chiesaestomba86@gmail.com (C.M.C.-E.); mohamad.khalife@epicura.be (M.K.); 2Department of Human Anatomy and Experimental Oncology, Faculty of Medicine, UMONS Research Institute for Health Sciences and Technology, University of Mons (UMons), B7000 Mons, Belgium; fabrice.journe@umons.ac.be; 3Department of Otolaryngology-Head & Neck Surgery, Foch Hospital, School of Medicine, UFR Simone Veil, Université Versailles Saint-Quentin-en-Yvelines (Paris Saclay University), F92150 Paris, France; 4Department of Otorhinolaryngology and Head and Neck Surgery, CHU de Bruxelles, CHU Saint-Pierre, School of Medicine, Université Libre de Bruxelles, B1000 Brussels, Belgium; 5Department of Surgery, EpiCURA Hospital, B7000 Hornu, Belgium; morgane.ducarme@gmail.com; 6Department of Internal Medicine, EpiCURA Hospital, B7000 Baudour, Belgium; sammy.place@epicura.be; 7Department of Otorhinolaryngology-Head & Neck Surgery, Hospital Universitario Donostia, 00685 San Sebastian, Spain; 8Department of Otolaryngology-Head & Neck Surgery, EpiCURA Hospital, B7000 Baudour, Belgium; 9Maxillofacial Surgery Unit, University Hospital of Sassari, 07100 Sassari, Italy; gderiu@uniss.it (G.D.R.); luigi.vaira@gmail.com (L.A.V.); 10Department of Intensive Care, EpiCURA Hospital, B7000 Hornu, Belgium; christophe.deterwangne@student.uclouvain.be (C.d.T.); shahram.machayekhi@epicura.be (S.M.); 11Institute for Medical Immunology, Université libre de Bruxelles, B1000 Brussels, Belgium; Arnaud.Marchant@ulb.be

**Keywords:** smell, olfactory, COVID-19, coronavirus, severe, anosmia

## Abstract

Objective: We investigate the prevalence of the self-reported and objective sudden loss of smell (SLS) in patients with severe coronavirus disease 2019 (COVID-19). Methods: Severe COVID-19 patients with self-reported SLS were recruited at hospitalization discharge. Epidemiological and clinical data were collected. The Sino-nasal Outcome Test-22 (SNOT-22) was used to evaluate rhinological complaints. Subjective olfactory and gustatory functions were assessed with the National Health and Nutrition Examination Survey (NHNES). Objective SLS was evaluated using psychophysical tests. Potential associations between olfactory evaluation and the clinical outcomes (duration of hospitalization; admission biology; one month serology (IgG), and chest computed tomography findings) were studied. Results: Forty-seven patients completed the study (25 females). Subjectively, eighteen (38.3%) individuals self-reported subjective partial or total SLS. Among them, only three and four were anosmic and hyposmic, respectively (38.9%). Considering the objective evaluation in the entire cohort, the prevalence of SLS was 21.3%. Elderly patients and those with diabetes had lower objective olfactory evaluation results than young and non-diabetic individuals. Conclusions: The prevalence of SLS in severe COVID-19 patients appears to be lower than previously estimated in mild-to-moderate COVID-19 forms. Future comparative studies are needed to explore the predictive value of SLS for COVID-19 severity.

## 1. Introduction

Since the onset of the coronavirus disease 2019 (COVID-19) pandemic, many patients reported sudden loss of smell (SLS) [1]. However, due to the health emergency, only a few studies investigated SLS with objective testing, which remains essential to confirm the olfactory dysfunction [2,3,4]. All these studies involved outpatients with mild COVID-19 forms. The mean age and the prevalence of comorbidities were low [2,3,4], leading some authors to suspect that SLS could be more specific to mild COVID-19 forms [5]. In this study, we investigated the prevalence of self-reported and objective SLS in severe COVID-19 patients. 

## 2. Methods

Adults (33–88 years old) with severe COVID-19 were recruited from the Department of Medicine of the EpiCURA Hospital (Hornu, Belgium, Ethics Committees: Epicura 2020–2303, Institut Jules Bordet CR 3180). The disease was confirmed through nasopharyngeal swab (RT-PCR). Patients were defined as severe COVID-19 if they required continuous care (oxygenotherapy, blood pressure monitoring) in internal medicine or intensive care units. 

Patients with a neurological disorder, chronic rhinosinusitis, or a history of nasal surgery prior to the pandemic were excluded. The lethal cases were not included because investigators assessed olfaction once the patient condition improved. Epidemiological and clinical data were collected at the hospital discharge. Details of the patient-reported outcome questionnaire used for data collection were reported in a previous study [3]. Briefly: (1) olfactory and gustatory questions were based on the smell and taste component of the National Health and Nutrition Examination Survey; (2) symptoms were evaluated through a 4 point scale ranging from 0 (no symptoms) to 4 (severe symptoms) [3]; (3) nasosinusal symptoms were evaluated through the French Sino-nasal Outcome Test-22 (SNOT-22) [6]. Patients benefited from psychophysical olfactory evaluation through sniffin’stick tests (Medisense, Groningen, The Netherlands): Sixteen pens were presented to patients every 30 s. The patient had to choose the adequate term describing the smell among 4 given options. The test was scored on a total of 16 points and allowed categorization into 3 groups: normosmia (score between 12 and 16), hyposmia (score between 9 and 11), and anosmia (score < 9) [3]. Moreover, the following hospitalization outcomes were recorded: duration of hospitalization (days); admission biology (D-dimer; hemoglobin; leucocyte count; lymphocyte count; CRP; creatitin; bilirubin; platelet count; LDH; Na^+^; K^+^; Cl^−^); 1 month serology (IgG); and chest computed tomography findings. Meanwhile, subjective and objective evaluations were made. 

The relationship between clinical and olfactory outcomes was analyzed through multiple linear regression between scale variables and through the Mann–Whitney test and boxplot representation for groups versus scale variables (SPSS, v22,0; IBM-Corp., Armonk, NY, USA). The local ethics committee approved the study (IJB:CE3137).

## 3. Results

Complete evaluation was performed in 47 patients, including 25 females. Patients were hospitalized in EpiCURA hospital from 20 March 2020, to 16 April 2020. Evaluations were conducted 41.0 ± 10.3 days after the onset of symptoms, corresponding to 1–2 weeks after the end of the hospitalization. Clinical outcomes are reported in Table 1. The most prevalent symptoms were: fever, asthenia, and anorexia. The mean duration of symptoms before hospitalization was 10.7 ± 5.0 days. Eight patients were hospitalized in the intensive care unit (ICU) for a mean duration of 8.5 ± 5.6 days. No patients received drugs for olfactory dysfunction. The CT scan and blood test features are reported in Table 1. 

Psychophysical olfactory evaluations indicated that four (8.5%) and nine (19.1%) patients reported anosmia and hyposmia (in the entire cohort), respectively (Table 2). Note that three hyposmic patients reported in the patient-reported outcome questionnaire that they had hyposmia prior to the infection. Excluding these three patients, the prevalence of objective SLS in our cohort was 21.3%. 

Eight and 10 patients experienced (subjective) total and partial loss of smell, respectively, over the clinical course of the disease; accounting for 38.3% of individuals. Among them, only three and four were anosmic and hyposmic (38.9%), respectively. The three patients who experienced hyposmia prior to the pandemic were not included in the subjective SLS patients. According to subjective evaluations of olfaction, thirty-eight-point-three percent of patients complained of SLS. Additional olfactory outcomes are reported in Table 2. 

Patients with diabetes had lower sniffin’stick test results compared with those without diabetes (Mann–Whitney U test; *p* = 0.045). The linear regression analyses revealed significant negative associations between the sniffin’stick test and age (r_s_ = −0.339; *p* = 0.032). Symptom duration was significantly correlated with the severity of fever (r_s_ = 0.395; *p* = 0.046) and dysphonia (r_s_ = 0.572; *p* = 0.002). Duration of hospitalization was significantly correlated with age (r_s_ = 0.402; *p* = 0.008). Serum IgG concentration measured by the SARS-CoV-2 LIAISON^®^ test (Diasorin, Centralino, Italy) was negatively correlated with the severity of nasal burning (r_s_ = −0.407; *p* = 0.029).

## 4. Discussion

Olfactory disorder is undoubtedly a key symptom of mild COVID-19 patients, affecting more than 70% of patients [4,5]. However, its prevalence remains uninvestigated in severe forms of the disease. In this study, we found that 38.3% of patients with severe disease experience SLS. Among them, thirty-eight-point-nine percent had abnormal objective tests one month after the onset of the infection. Irrespective of the method used to evaluate the prevalence of SLS (patient-reported outcome questionnaire versus objective tests), these data indicate that SLS could be more prevalent in mild-to-moderate forms of the infection. 

According to a previous study conducted in the same population and with the same methods, self-reported SLS concerned more than 70% of mild COVID-19 patients, and among them, sixty-two percent had abnormal objective evaluations [3]. The higher incidence of SLS in mild forms of COVID-19 suggests a relative compartmentalization of the disease. Such compartmentalization may involve differences in immune responses to SARS-CoV-2 at the level of nasal and olfactory mucosa. In patients with potent mucosal immune responses, viral replication and dissemination to the lower respiratory tract may be better controlled, and this could be at the expense of local inflammation and symptoms involving nasal and bulb regions. In patients with less potent mucosal immune responses, viral replication could spread to the lower respiratory tract and lead to systemic immune response and inflammation. This hypothesis is supported by our observation that nasal burning was inversely correlated with SARS-CoV-2 serum IgG, whereas severe forms of the disease have been positively correlated with SARS-CoV-2 IgG responses [7]. Further studies are needed to test this hypothesis. Both age and diabetes could be favoring factors in the development of SLS, which is well known in other olfactory diseases [8,9]. The mechanisms underlying the development of olfactory dysfunction in patients with diabetes may involve neuropathy and damage in the olfactory nerves. 

The main limitations of the present study are the low number of patients, the lack of a control group, and the performance of olfactory tests one month after the onset of symptoms. Performing the tests during hospitalization was difficult due to the sanitary situation, the patient clinical state, and the difficulties in correctly sensing the pens with transnasal oxygenation. Although this possibility is not supported by patient-reported symptoms, the delay between the onset of symptoms and the objective olfactory testing may underestimate the incidence of olfactory dysfunction because of olfactory mucosa recovery.

## 5. Conclusions

The prevalence of SLS in severe COVID-19 patients appears to be lower than previously estimated for mild-to-moderate COVID-19 forms. Future comparative studies are needed to explore the predictive value of SLS for COVID-19 severity. 

## Figures and Tables

**Table 1 pathogens-09-00627-t001:** Epidemiological and clinical characteristics of patients.

Characteristics	Patients (N-%)
Age (mean-SD) (y)	58.8 ± 12.9
Gender (female/male)	25/22
Ethnicity	
Caucasian	44 (93.6)
North African	2 (4.3)
Black African	1 (2.1)
Smoker	0 (0)
Patients with seasonal allergy	12 (25.5)
Comorbidities	
Hypertension	10 (21.3)
GERD	9 (19.1)
Hypothyroidism	9 (19.1)
Diabetes	7 (14.9)
Asthma	5 (10.6)
Heart problems	4 (8.5)
Neurological diseases	3 (6.4)
Renal failure	2 (4.2)
Hepatic insufficiency	1 (2.1)
Untreated cancer	1 (2.1)
Depression	1 (2.1)
Autoimmune disease	0 (0)
Respiratory insufficiency	0 (0)
General Symptoms	
Asthenia	44 (93.6)
Fever (>38 ℃)	44 (93.6)
Anorexia	44 (93.6)
Dyspnea	41 (87.2)
Cough	38 (80.9)
Myalgia	36 (76.6)
Headache	35 (74.5)
Diarrhea	32 (68.1)
Arthralgia	27 (57.4)
Chest pain	26 (55.3)
Nausea/vomiting	24 (51.1)
Abdominal pain	22 (46.8)
Conjunctivitis	14 (29.8)
Ear, Nose, and Throat Symptoms	
Rhinorrhea	33 (70.2)
Nasal obstruction	30 (63.8)
Dysphonia	27 (57.4)
Throat sputum	26 (55.3)
Postnasal drip	25 (53.2)
Sore throat	23 (48.9)
Dysphagia	21 (44.7)
Face pain/heaviness	18 (38.3)
Nose burning	15 (31.9)
Ear pain	14 (29.8)
Presumed hyposmia	10 (21.3)
Presumed anosmia	8 (17.2)
Cacosmia	8 (17.2)
Taste dysfunction	6 (12.8)
Phantosmia	1 (2.1)
Hospitalization Findings	
ICU patients	8 (17.0)
Duration of symptoms before hospitalization (mean, SD)	10.7 ± 5.0
Hospitalization duration (mean, SD-range, days)	8.7 ± 4.8 (2–21)
Chest CT-Scan Findings (Lung Involvement)	
10–25%	9 (19.1)
25–50%	23 (48.9)
>50%	6 (12.8)
>75%	1 (2.1)
Missing data	8 (17.0)
Biology Features	
Hemoglobin (g/dL)	14.0 ± 1.6
Neutrophils (10^3^/µL)	6.8 ± 3.4
Lymphocytes (10^3^/µL)	1.1 ± 0.5
*Lymphopenia*	34 (72.3)
*Normopenia*	13 (27.7)
Platelets (10^3^/µL)	242.9 ± 113.2
CRP (mg/L)	119.5 ± 110.1
Creatinine (mg/dL)	1.1 ± 0.8
Bilirubin (mg/dL)	0.5 ± 0.3
D-dimer (µg/L)	1258.0 ± 531.1
LDH (UI/L)	362.4 ± 138.3
Na^+^ (mmol/L)	136.9 ± 3.6
K^+^ (mmol/L)	4.1 ± 0.7
Cl^−^ (mmol/L)	97.2 ± 4.1
1 month mean (SD) IgG level	173.3 ± 80.6

Abbreviations: CRP = C-reactive protein; CT = computed tomography; GERD = gastroesophageal reflux disease; SD = standard deviation.

**Table 2 pathogens-09-00627-t002:** Olfactory outcomes.

Olfactory Outcomes	
Aroma Perception Disorder	N = 12
Total vs. partial loss of aroma perception sense	1 (2.1)/6 (12.8)
Distortion	5 (10.6)
Olfactory Outcomes	
Variable olfactory dysfunction	8 (44.4)
Nasal obstruction related dysfunction	5 (27.8)
Non-variable	3 (16.7)
Did not remember	2 (11.1)
Onset of Smell Dysfunction	N = 18
Before the other symptoms	1 (5.6)
Concomitant with other symptoms	9 (50.0)
After the other symptoms	8 (44.4)
Did not remember	0 (0)
Sniffin’sticks Tests (Mean, SD)	N = 47
Mean value	12.7 ± 2.8
Anosmic	4 (8.5)
Hyposmic	9 (19.1)
Normosmic	34 (72.3)
SNOT-22 (Mean, SD)	41.1 ± 18.6

The number in parentheses represents the percentage. Abbreviations: SD = standard deviation; SNOT-22= Sino-nasal Outcome Test-22 questionnaire.

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
