# Peer review of "Objective Olfactory Findings in Hospitalized Severe COVID-19 Patients"

_pathogens, 2020, doi:10.3390/pathogens9080627_

Round 1

Reviewer 1 Report

The authors provide data on olfactory dysfunction in severe COVID-19 patients. They distinguish between self-reported symptoms and objective findings in well-recognized tests and correlate their findings with comorbidities. 

The present study is of interest and importance. The manuscript is well-written and methods as well as results are described and presented well. The authors are able to name the limitations of their study which is highly appreciated. However, few aspects should be addressed to improve the overall relevance of their work.

  1. What about the COVID-19 cases with lethal outcome? The authors are encouraged to cite the respective literature. Does this fit to the present results: the more severe the less loss of smell?
  2. The authors discuss the limitation of their results which are, in parts, due to objective testing when patients were discharged from hospital. The authors could discuss why this might be a problem -> regeneration of the olfactory epithelium, limited life-span of olf. neurons etc.
  3. What about gustatory dysfunctions? Were any tests performed on this aspect?
  4. Could the authors speculate why the score of the sniffing' stick test was lower in patients with diabetes? Is this score usually comparable in non-infected diabetes patients vs. healthy individuals? Thinking of diabetic polyneuropathy and pre-damaged neurons- is there anything known?
  5. Minor comment: Table 1: should be conjunctivitis.

Author Response

  1. What about the COVID-19 cases with lethal outcome? The authors are encouraged to cite the respective literature. Does this fit to the present results: the more severe the less loss of smell?

We specified : methods, line 4 : « The lethal cases were not included because investigators assessed olfaction once the patient condition was improved. »

  1. The authors discuss the limitation of their results which are, in parts, due to objective testing when patients were discharged from hospital. The authors could discuss why this might be a problem -> regeneration of the olfactory epithelium, limited life-span of olf. neurons etc.

We specified : last line of the manuscript : « Although this possibility is not supported by patient-reported symptoms, the delay between the onset of symptoms and the objective olfactory testing may underestimate the incidence of olfactory dysfunction because olfactory mucosa recovery. »

  1. What about gustatory dysfunctions? Were any tests performed on this aspect?

No objective evaluation. Just subjective assessment (using the taste component of the National Health and Nutrition Examination Survey). We specified the taste dysfunction rate in table 1.

  1. Could the authors speculate why the score of the sniffing' stick test was lower in patients with diabetes? Is this score usually comparable in non-infected diabetes patients vs. healthy individuals? Thinking of diabetic polyneuropathy and pre-damaged neurons- is there anything known?

As requested, we specific : discussion, line 12 : « Further studies are needed to test this hypothesis. Both age and diabetes could be favoring factors in the development of SLS, which is well known in other olfactory diseases [8,9]. The mechanisms underlying the development of olfactory dysfunction in patients with diabetes may involve neuropathy and damage in the olfactory nerves. »

  1. Minor comment: Table 1: should be conjunctivitis.

It’s corrected.

Reviewer 2 Report

Brief summary

The brief report entitled “Objective Olfactory Findings in Hospitalized Severe COVID-19 Patients” edited by Lechien and colleagues aimed to investigate the sudden loss of smell (SLS) in patients with severe COVID-19. To evaluate this clinical consequence the authors administered a questionnaire to 47 patients, and they used psychophysical tests at the moment of hospitalization discharge.

Broad comments

The work is original, well written and it is very interesting. The research is useful to evaluate the possible clinical consequences of COVID-19 on the olfactive system. This is preliminary research that requires further in-depth studies. The findings could suggest a research line for the future.

Specific comments

  • I would make a small question: Is there a follow-up scheduled for the same patients six months after discharge?
  • At the time of the study, were patients taking drug therapy?
  • Table 2. About aroma perception disorder, olfactory outcomes, and onset of smell dysfunction, does the number in parentheses indicate the standard deviation? Please, specify it.
  • I suggest replacing "the health emergency" with "health emergency". Following the studies conducted on the COVID-19 pandemic and with the conceptualizations of the WHO, the term “health” is more appropriate than “sanitary”.
  • I suggest to add the absence of a control group as a methodological limitation of the study.

Author Response

I would make a small question: Is there a follow-up scheduled for the same patients six months after discharge?

Yes. A follow-up is made through phone call (for the clinical condition) and the anosmic patients will be follow-up in ENT consultation.

At the time of the study, were patients taking drug therapy?

They received drugs in relation to their clinical condition (ICU unit,, etc.) but no drugs for their olfactory disorder. We specified that in the results as suggested: results, line 4 « No patient received drugs for the olfactory dysfunction. »

Table 2. About aroma perception disorder, olfactory outcomes, and onset of smell dysfunction, does the number in parentheses indicate the standard deviation? Please, specify it.

No, it’s the percentage. We specified : « The number in parentheses represents the percentage. »

I suggest replacing "the health emergency" with "health emergency". Following the studies conducted on the COVID-19 pandemic and with the conceptualizations of the WHO, the term “health” is more appropriate than “sanitary”.

Done.

I suggest to add the absence of a control group as a methodological limitation of the study.

We specified : discussion, last paragraph : « The main limitations of the present study are the low number of patients, the lack of control group and the performance of olfactory tests one month after the onset of symptoms. »
